# Residue Concentrations of Cloxacillin in Milk after Intramammary Dry Cow Treatment Considering Dry Period Length

**DOI:** 10.3390/ani13162558

**Published:** 2023-08-08

**Authors:** Carola Fischer-Tenhagen, Detlev Bohm, Anke Finnah, Sebastian Arlt, Samira Schlesinger, Stefan Borchardt, Franziska Sutter, Christie M. Tippenhauer, Wolfgang Heuwieser, Peter L. Venjakob

**Affiliations:** 1Clinic for Animal Reproduction, Faculty of Veterinary Medicine, Freie Universität Berlin, Königsweg 65, 14163 Berlin, Germany; sebastian.arlt@fu-berlin.de (S.A.);; 2Center for Protection of Experimental Animals, German Federal Institute for Risk Assessment (BfR), Alt Marienfelde 17-21, 12277 Berlin, Germany; 3Federal Office of Consumer Protection and Food Safety (BVL), Gerichtstraße 49, 13347 Berlin, Germany; 4Clinic for Ruminants, Faculty of Veterinary Medicine, Justus-Liebig-Universität Giessen, Frankfurter Str. 104, 35392 Giessen, Germany

**Keywords:** dry off, antibiotic residues, short dry period

## Abstract

**Simple Summary:**

In the dairy industry, cows are dried off approximately six weeks before calving. This is to regenerate the udder tissue and to cure potential infections in the udder with the help of antibiotic dry cow treatment. The dry period can be shortened intentionally, but also accidentally in the case of premature calving. A shortened dry period bears the risk of antibiotic residues in milk. In this study, we evaluated cloxacillin concentrations in milk of individual udder quarters treated 6 to 32 d before calving. Even with intervals as short as 6 d, concentrations of cloxacillin were below the maximum residue limit of 30 µg/kg at 5 d after calving.

**Abstract:**

Dry cow treatment with an intramammary antibiotic is recommended to reduce the risk of mastitis at the beginning of the next lactation. The dry period may be shortened unintentionally, affecting antibiotic residue depletion and the time when residues reach concentrations below the maximum residue limit (MRL). The objective of this study was to evaluate residue depletion in milk after dry cow treatment with cloxacillin, considering dry periods of 14 (G14d), 21 (G21d), and 28 d (G28d). Overall, fifteen cows with 60 udder quarters were included in the study. For each cow, three of the udder quarters were treated with 1000 mg cloxacillin benzathine (2:1) on d 252, d 259, and d 266 of gestation; one quarter was left untreated. Milk samples were drawn until 20 DIM and milk composition, somatic cell count and cloxacillin residues were analyzed. The HPLC-MS/MS revealed different excretion kinetics for the compounds cloxacillin and cloxacillin benzathine (1:1). All cows showed a cloxacillin and cloxacillin benzathine (1:1) concentration below the MRL of 30 µg/kg after 5 d. In the udder quarters of G21d and G28d, the cloxacillin concentration was already below the MRL at first milking after calving. The cloxacillin benzathine (1:1) concentration in the milk of G28d, G21d, and G14d fell below 30 µg/kg on the 5th, 3rd, and 5th DIM, respectively. Shortening the dry period affects residue depletion after dry cow treatment with cloxacillin. The risk of exceeding the MRL, however, seems low, even with dry periods shorter than 14 d.

## 1. Introduction

The dry period is essential for mammary cell renewal to improve milk production in the following lactation and to improve the cure rate in cows with subclinical mastitis [1]. The dry period comprises the last six to eight weeks of gestation [2,3]. Cows with a high somatic cell count at dry-off, or a history of mastitis in the last lactation are at increased risk of developing mastitis during the dry period or at the onset of the next lactation [4]. Accordingly, dry cow treatment (DCT) with an intramammary antibiotic or teat sealant is recommended as part of effective dry cow management [5] and reduces the risk of mastitis at the beginning of the next lactation [6,7,8]. In the US, the average dry period length is 57 d with 93% of dry cows receiving an intramammary dry cow antibiotic [9]. A survey conducted in Northern Germany revealed an average dry period length of approximately 50 d with about 80% of farms using a blanket DCT [10].

In the dairy industry, mastitis control is one of the major contributors to antibiotic use [11,12]. Of the antibiotics used for this purpose, 73% are DCT and only 27% are used for treatment of clinical mastitis [13]. One concern of antibiotic treatment in food-producing animals, particularly of antibiotic DCT, is the risk of antibiotic residues in milk [14]. Shortening the dry period may lead to a higher risk of antibiotic residues in milk beyond the withdrawal period [15].

There are various pros and cons for both short and long dry periods. Shortening the dry period reduces milk production in the early stages of the next lactation while similar or increased feed intake is observed. This can improve the energy balance and metabolic status of the cow [16,17]. However, Andrée O’Hara et al. [18] found that a 4-week dry period was associated with a greater incidence of mastitis in the following lactation than an 8-week dry period and [19] stated that colostrum production was compromised in cows with a short dry period. In addition, a large number of studies have demonstrated the negative effect of shortening or omitting the dry period on milk production during the following lactation [20].

Cloxacillin is a semisynthetic penicillin, well known as an effective drug against *Staphylococcus aureus*. With a prevalence of 11% in mastitis cases, *Staphylococcus aureus* is the second most prevalent Gram-positive pathogen identified in cows with mastitis in Germany [21]. Given the persistent udder infections associated with this pathogen, antibiotic DCT is of particular importance [22]. In Germany, cloxacillin is commonly used in mastitis control. Currently, five different products are approved for the treatment at dry off and three different products are approved for the treatment of clinical mastitis. In Europe, it is generally not allowed to put milk on the market that was milked during the first 5 DIM. The European Commission has set maximum residue limits (MRL) for drug residues in food. According to Commission Regulation (EU) No. 37/2010 [23], the MRL for cloxacillin is 30 µg/kg of milk. The US Food and Drug Association version of MRL is tolerance, (i.e., 10 µg/kg for cloxacillin [24]). While cloxacillin is commonly used for antibiotic DCT in the dairy industry [25,26] there are only a few in vivo studies [27,28] addressing the risk of cloxacillin residues in milk after calving. The objective of the present study was to determine the effect of different dry cow periods after DCT on cloxacillin residue concentrations in milk after calving. Our hypothesis was that cloxacillin concentrations in milk would exceed the MRL for a prolonged period of time if a short dry period (i.e., <21 d) is used.

## 2. Materials and Methods

### 2.1. Animals, Housing and Eligibility Criteria

The study was conducted between April 2018 and May 2020 at the Clinic for Animal Reproduction, Freie Universität Berlin. The experimental procedures reported herein were conducted with the approval of the federal authorities (protocol A 0363/17). In accordance with previous studies, udder quarters of each cow were considered as independent units, which permitted a reduction of the number of study animals [29,30,31]. An a priori sample size calculation was conducted using G*Power (version 3.1.9.2; University of Düsseldorf, Düsseldorf, Germany). Using α of 5%, a power of 95% and an effect size of 0.6, 45 experimental units (15 udder quarters per treatment group) were needed to demonstrate a statistically significant difference between cloxacillin concentrations in the milk of the treated groups. Cows enrolled in this study were purchased from a commercial dairy farm in northern Germany, housing approximately 2600 lactating cows with an average 30 d milk yield of 9600 kg. Due to the comprehensive sampling scheme, the study was conducted in 3 consecutive replicates. While 5 cows were enrolled in the 1st replicate, 6 cows were enrolled in each of the remaining 2 replicates. Cows were eligible for inclusion if they were (1) Holstein Friesian cows, (2) at the end of their 1st lactation, around 1 week before dry off, (3) had no case of clinical mastitis during their 1st lactation, (4) had a somatic cell count below 100,000 cells/mL in their last 3 dairy herd improvement association (DHIA) equivalent tests (monthly test for milk yield, milk components and somatic cell count), and (5) did not receive antibiotic DCT before the start of the study. Approximately, 9 weeks before expected parturition, cows were transferred from the farm to the Clinic for Animal Reproduction in order to ensure that DCT was not applied on the farm (usually conducted 8 weeks before expected parturition). Cows were dried off upon arrival at the Clinic for Animal Reproduction. All cows were managed according to the guidelines set by the International Cooperation on Harmonization of Technical Requirements for Registration of Veterinary Medicinal Products [32]. During the study period, cows were housed in a freestall barn with 6 cubicles equipped with chopped straw–lime–water mixture and concrete flooring. All cows were fed grass silage, haylage, and hay ad libitum supplemented with standard dairy concentrate (Milchleistungsfutter MLF 18/3 Standard; BKF Belziger Kraftfutter GmbH, Bad Belzig, Germany) according to pregnancy status and milk yield. Fresh water was available ad libitum. Cows were monitored daily for signs of imminent calving.

### 2.2. Prepartum Study Protocol

Before the transfer from the farm to the Clinic for Animal Reproduction, all cows were carefully examined to assure that the animals met the inclusion criteria. Apart from the eligibility criteria mentioned above, a clinical examination (i.e., respiration rate, body temperature, heart rate, rumen contraction), a trans-rectal pregnancy check, a visual udder examination (i.e., rubor and tumor), a thorough udder palpation (i.e., tumor, calor, and dolor), an evaluation of hyperkeratosis (i.e., no ring, smooth or slightly rough ring, rough ring, and very rough ring; [33]), and a classification of the shape of the teat ends (i.e., pointed, inverted, plate, and round; [34]) was performed. All cows were examined by the last author.

The treatment protocol was initiated using an antibiotic DCT consisting of 1000 mg cloxacillin benzathine 2:1 (Cloxacillin-Benzathin 1000 mg T.S., CP-Pharma, Burgdorf, Germany; compound consisting of 2 molecules cloxacillin and 1 molecule of benzathine; authorization number 6232.00.00). For each cow, 3 of the udder quarters received the antibiotic DCT on d 252, d 259, and d 266 of gestation, respectively. The 4th quarter was left untreated. The allocation of udder quarter to treatment group occurred based on a random allocation list created in Excel (Office 2019, Microsoft Deutschland Ltd., Munich, Germany). Due to the dry-off 9 weeks before expected parturition, udder involution was complete before the 1st DCT on d 252 of gestation. All treatments were conducted at 1300 h by the same person (last author).

Application of DCT was conducted according to manufacturer guidelines. In brief, teat ends were cleaned with paper towels soaked with 70% alcohol until they were completely clean. The teat cistern was compressed using 2 fingers, the tip of the intramammary tube was introduced into the teat canal, the intramammary DCT was injected, and massaged upwards to promote dispersion within the glandular tissue. After the 1st treatment at d 252 of gestation, cows were examined daily at 07:00 h. This examination comprised auscultation of the rumen, measurement of the rectal temperature, visual assessment of the udder (i.e., milk leakage, swelling, and redness), and visual assessment of signs for imminent calving (i.e., edema and discharge from the vulva and edema of the udder). Milk leakage was defined as milk dripping or flowing from the teat.

When the amniotic sac or the calf’s feet were visible outside the vulva, animals were moved to an individual maternity pen (3.50 × 3.10 m) equipped with rubber mats and deep straw bedding, separated from the study pen by 2 metal fences. Immediately after birth, the calf was separated from the dam and the cow was transferred back to the study pen.

### 2.3. Postpartum Study Protocol

Until 20 DIM cows were milked twice daily at 07:00 h and 19:00 h, using a portable bucket milking machine (i.e., GEA milking bucket with pulsator Vacupuls Constant; GEA Westfalia Separator AG, Oelde, Germany). A vacuum line connected the milking bucket to an RPS Vacuum Pump (800 L/min) equipped with a 3-phase motor providing a typical milking cluster (Classic 300 milking cluster; GEA Westfalia Separator AG). Prior to milking, each teat was wiped using an individual wet paper towel followed by an individual dry paper towel. The first 3 streams of milk were milked into a pre-milking cup, evaluated for abnormal milk character (i.e., clots, flakes, discoloration, traces of blood, or abnormal consistency of secretions) and subsequently discarded. A clinical examination of each cow (general demeanor, heart rate, breathing rate, rectal temperature, and rumen contractions) was performed and documented once daily, before the morning milking.

### 2.4. Sample Collection and Analysis

At each milking, 2 foremilk samples (i.e., 10 mL and 40 mL) were collected immediately after the pre-milking procedure. The 40 mL foremilk samples were stored at 6 °C and transferred biweekly to the local DHIA testing laboratory (Landeskontrollverband Berlin-Brandenburg e.V., Waldsieversdorf, Germany). At the laboratory, somatic cell count (SCC) was determined using flow cytometry according to IDF 148–2:2006 [35] and fat, protein and lactose were measured using mid-infrared spectroscopy [36]. Urea was determined using continuous flow analysis according to the guideline of the German Association for Performance and Quality testing [36].

The 10 mL foremilk samples were frozen at −80 °C and analyzed after sample collection was completed. Residue concentrations of the relevant analytes cloxacillin and cloxacillin benzathine (1:1) were determined by muva Kempten GmbH (Kempten, Gemany). During preliminary investigations of study samples, a considerable signal in the chromatograms was observed. The signal was identified as cloxacillin benzathine (1:1) by comparison with a reference standard and shows another retention time than cloxacillin in the chromatogram. Cloxacillin benzathine (1:1) is a compound in which 1 molecule of cloxacillin is linked to 1 molecule of benzathine. Thus, we decided to analyze cloxacillin and cloxacillin benzathine (1:1) separately.

The first 21 foremilk samples (i.e., 1st to 10th DIM) from each cow were analyzed consecutively for both compounds until the concentration reached a 10th of the MRL (i.e., 3 µg/kg). In 58/60 udder quarters, the concentration was below this limit on the 9th milking. In the remaining 2 cows, the concentration fell below this limit by the 11th milking. Hence, we restricted our results to the 1st, 2nd, 3rd, 5th, 7th, and 9th milking.

The thawed samples were homogenized for 10 s using a vortexer (Heidolph, Kelheim, Germany). An analytical sample of 1 g of foremilk was weighed into a centrifuge tube (PE tube, 15 mL, Sarstedt, Nürnbrecht, Germany). Then the internal standard dicloxacillin was added to the analytical sample. For protein separation, 6 mL of acetonitrile were added. The mixture was carefully vortexed for 10 s and shaken for 10 min. Subsequently, the mixture was centrifuged (3800× *g*, 5 min, 5 °C, Heraeus, Hanau, Germany) and the supernatant of acetonitrile was collected in a tube (PE tube, 12 mL, Greiner, Kremsmünster, Austria). The acetonitrile phase was evaporated to dryness under a weak nitrogen stream in a TurboVap evaporator (Zymark, Idstein, Germany) at 40 °C. Then, the residue was reconstituted in 5 mL of buffer solution (Mix of 0.1 M citric acid and 0.2 M Na_2_HPO_4_ * 2 H_2_O with a ratio 60/40 *v*/*v* and added 0.1 M Na_2_EDTA * 2 H_2_O). The clean-up of this extract fluid containing the analytes was performed by solid-phase extraction. The OASIS HLB cartridge (200 mg, 6 mL, Waters, Eschborn, Germany) was positioned on the vacuum station (Supelco, Taufkirchen, Germany) and conditioned with 6 mL of methanol followed by 6 mL of water. Then the extracted fluid of buffer (5 mL) was applied to the cartridge. The cartridge was washed with 6 mL of a 95/5 *v*/*v* water/methanol mixture and dried for 20 min in an air stream. After the drying step, 6 mL of methanol was added to elute the analytes into a tube (PE tube, 12 mL). The eluate was evaporated to dryness under a weak nitrogen stream in a TurboVap evaporator at 40 °C. Then the residue was reconstituted in a 1.0 mL mix of water/acetonitrile 90/10, *v*/*v*, centrifuged and the supernatant was transferred into LC vial (WICOM, Heppenheim, Germany). A volume of 10 µL of this solution was injected into the LC-MS/MS system. The measurement system consisted of an HPLC 1290 (Agilent Technologies, Waldbronn, Germany) containing an LC column (Phenomenx “Aqua”, 150 × 2 mm, 3 µm, Aschaffenburg, Germany) and coupled with the MS/MS detector 6500+ (SCIEX, Darmstadt, Germany). A mobile phase of 0.1% formic acid and acetonitrile containing 0.1% formic in gradient mode at an oven temperature of 30 °C and 0.3 mL/min flow was used. The MS/MS measurement in triple quadrupole mode was performed using ionization ESI+ and scan-type multiple reactions monitoring MRM. To quantify and qualify the analytes, the MRM transition 436/277,160 was used for cloxacillin, 676/410,517 for cloxacillin benzathine (1:1) and 470/160,311 for the internal standard dicloxacillin (Figure 1). The quantification was conducted on the basis of matrix calibration curves with the internal standard dicloxacillin. A limit of quantification (LOQ) of 0.1 µg/kg was determined for cloxacillin and cloxacillin benzathine (1:1) each. The precision of this method was 5.1% and 6.6% for cloxacillin and cloxacillin benzathine (1:1), respectively. The accuracy was 100.0% and 96.4% for cloxacillin and cloxacillin benzathine (1:1). This method can be applied to cloxacillin measurement in a concentration range around the MRL of 30 µg/kg and cloxacillin benzathine (1:1) measurement at higher concentrations.

### 2.5. Statistical Analysis

Individual quarter data and results from sample analyses were entered into Excel spreadsheets (Office 2010) and analyzed using SPSS for Windows (version 25.0, IBM Corp., Armonk, NY, USA). For statistical analysis, the udder quarters were grouped according to the interval from treatment to calving: 6 to 17 d (G14d; *n* = 17), 18 to 24 d (G21d, *n* = 15) and 25 to 32 d (G28d; *n* = 13). To analyze the residue concentrations of cloxacillin benzathine (1:1) in milk a repeated measures ANOVA with first-order autoregressive covariance was performed (GENLINMIXED procedure of SPSS). The milking number was used as repeated measure. In order to obtain a normal distribution, residue concentrations of the antibiotic were transformed using a logarithmic scale. Udder quarter within cow was used as a random effect. According to the model-building strategies described by Dohoo et al. (2009), each parameter considered for the mixed model was separately analyzed in a univariable model. Only parameters resulting in univariable models with *p* ≤ 0.1 were included in the final mixed model. Selection of the model that best fits the data was performed by using a backward stepwise elimination procedure that removed all variables with *p* > 0.1 from the model. The initial model contained the following explanatory variables: Interval between treatment and calving (G14d vs. G21d vs. G28d), milking number relative to calving (i.e., 1st, 2nd, 3rd, 5th, 7th, and 9th milking after calving), fat content in milk (%, continuous), protein content in milk (%, continuous), lactose content in milk (%, continuous), somatic cell count transformed to the linear score ([37]; continuous) and milk yield per milking (kg, continuous). Model diagnostics included visual inspection of normality and homoscedasticity of residuals. Regardless of the significance level, treatment, the number of milking relative to calving and the interaction of number of milking by treatment were forced to remain in the model.

Results on cloxacillin residue concentrations were not normally distributed. Because 353 of 376 samples analyzed for cloxacillin had a concentration < 10 µg/kg, a transformation did not result in normal distribution. Therefore, we refrained from using repeated measures ANOVA and presented the results descriptively as mean, median, and range stratified by treatment group and milking number relative to calving (Table 1). To evaluate the number of milkings needed for cloxacillin and cloxacillin benzathine (1:1) concentration to fall below 30 µg/kg, a Kaplan–Meier survival analysis was performed. Cows were stratified based on the interval between dry cow treatment and calving (G14d vs. G21d vs. G28d) and were censored if the concentration of cloxacillin or cloxacillin benzathine (1:1) fell below the concentration of 30 µg/kg.

## 3. Results

Overall, 17 Holstein Friesian cows were enrolled in this study. A total of two cows had to be excluded; one due to milk leakage a few days before calving and another due to abortion on d 253 of gestation. The remaining fifteen cows (60 quarters) were included in the final statistical analysis. The average interval from DCT to calving was 12.5, 20.4, and 27.5 d for G14d (*n* = 17), G21d (*n* = 15), and G28d (*n* = 13), respectively.

### 3.1. Analytics

The HPLC-MS/MS revealed a peak at 9.79 min and a second peak at 12.19 min. Using cloxacillin and cloxacillin benzathine (1:1) spiked calibration samples, it was confirmed that the peak detected at 9.79 min referred to cloxacillin benzathine (1:1) and the peak detected at 12.19 min referred to cloxacillin. A limit of quantification (LOQ) of 0.1 µg/kg was determined for cloxacillin and cloxacillin benzathine (1:1) each. As we observed different excretion kinetics for cloxacillin and cloxacillin benzathine (1:1), results are reported separately.

### 3.2. Residue Depletion

After 5 d (11th milking after calving), the concentrations of cloxacillin and cloxacillin benzathine (1:1) in milk were below a 10th of the MRL (i.e., 3 µg/kg) in all samples. Therefore, no further milk samples were analyzed. Cloxacillin: In the udder quarters of G21d and G28d, the concentration of cloxacillin was already below the MRL at first milking after calving. In G21d and G28d, however, a concentration above the MRL was detected on d 3 (58.15 µg/kg), and d 2 (34.70 µg/kg), respectively (Table 1). In G14d, cloxacillin excretion varied between udder quarters. While cloxacillin concentrations were below the MRL at first milking in 13 of 17 milk samples, in each of two milk samples the concentration was below the MRL at the fifth and tenth milking, respectively.

Cloxacillin benzathine (1:1): In G28d, the concentrations of cloxacillin benzathine (1:1) in the milk fell below 30 µg/kg at the ninth milking after calving at the latest (5 DIM). In G21d, the concentration of cloxacillin benzathine (1:1) in the milk of all 15 udder quarters was below 30 µg/kg of milk at the fifth milking after calving (3 DIM). At the ninth milking, there was a concentration above the MRL in one cow (36.81 µg/kg). In G14d, no cloxacillin benzathine (1:1) concentrations above 30 µg/kg were detected in any udder quarter after the ninth milking. The mean, median, and range of cloxacillin benzathine (1:1) removal in milk are displayed in Table 2.

### 3.3. Effect of Dry Period Length

Cloxacillin: Mean, median, and range of cloxacillin removal in milk are displayed in Table 1. In G14d cows, mean and median cloxacillin concentrations clearly show that cloxacillin residues decrease continuously with the number of milkings. As indicated by the range of cloxacillin concentrations, there were, however, considerable animal-specific differences in cloxacillin depletion. In G21d and G28d, cloxacillin concentrations were comparatively low from the first milking after calving. Interestingly, despite initially low concentrations, some cows in these two groups showed moderately elevated cloxacillin concentrations at a later milking (i.e., third, seventh, ninth milking), albeit detected concentrations later than the third milking were still below the MRL.

Cloxacillin benzathine (1:1): The number of milkings relative to calving (*p* < 0.001) and the d of DCT (*p* < 0.001) had a significant effect on cloxacillin benzathine (1:1) excretion. Shorter treatment-calving intervals were associated with greater concentrations of cloxacillin benzathine (1:1) in milk (Figure 2). Beyond the 7th milking, however, no differences were detected. Moreover, lactose (*p* < 0.001) and protein (*p* = 0.008) content affected the cloxacillin benzathine (1:1) excretion. A 1% increase in protein and lactose was associated with a 1.1 µg/kg and a 0.4 µg/kg increase of cloxacillin benzathine (1:1) in milk, respectively. Table 2 also shows that some cows have increased cloxacillin benzathine (1:1) concentrations at later milkings despite initially low cloxacillin–benzathine (1:1) concentrations.

### 3.4. Untreated Udder Quarters

Residues of cloxacillin and cloxacillin benzathine (1:1) were also found in the untreated udder quarters (Table 1 and Table 2). At first milking, mean concentrations of cloxacillin and cloxacillin benzathine (1:1) were 0.01 ± 0.02 µg/kg of milk (range 0.0 to 0.10) and 0.22 ± 0.31 µg/kg of milk (range 0.0 to 1.1 µg/kg), respectively.

## 4. Discussion

To prevent antibiotic residues above the MRL in the milk after calving, withdrawal periods for milk are determined using the time-to-safe-concentration method, which calculates a tolerance limit on the number of milkings per animal. This tolerance limit is the time necessary for the residue concentration in the milk of the majority of animals to fall below the safe threshold (i.e., the MRL; [38]). In order to determine withdrawal periods for DCT, it is recommended to use cohorts of animals with as little variance in dry period length as possible. The number of d, which must pass before the milk is deemed safe for human consumption, is called the withdrawal period. For products for DCT the length of the preceding dry period needs to be specified in addition. For the cloxacillin product used in this study, a withdrawal period of 5 d after calving for cows treated more than 35 d before calving, and 40 d for cows treated less than 35 d before calving was set by the competent authority. This approach for defining the withdrawal period is based on the fact that antibiotic residues in early lactation have been shown to differ if the dry period is cut short due to early calving [15].

We treated cows with the approved dose of 1000 mg of cloxacillin–benzathin (2:1) on d 252, d 259, and d 266 of gestation, resulting in a DCT to a calving interval of 6 to 32 d. At 5 DIM, cloxacillin concentrations were below the MRL of 30 µg/kg in all milk samples, even after a dry period of only 6 d. These implausibly low concentrations of cloxacillin were detected in a preliminary analysis based on the reference method for confirmation of residues of cloxacillin in milk [39]. However, the HPLC-MS/MS revealed two peaks one at 9.79 min and one at 12.19 min (Figure 1). The chromatograms of analyzed milk samples showed the typical MRM transitions for cloxacillin (12.19 min) and cloxacillin benzathine (1:1, 9.79 min). We postulated that after treatment with cloxacillin benzathine (2:1), the compounds cloxacillin and cloxacillin benzathine (1:1) were formed in milk. The use of standard material of both compounds allowed the quantification on the basis of matrix calibration curves supported by the internal standard dicloxacillin and underlined the postulation. This finding was confirmed by creating chromatograms of cloxacillin benzathine (1:1) spiked calibration samples. The antimicrobial activity of cloxacillin benzathine (1:1) is not proven. Preliminary results on antimicrobial inhibitor tests suggest that it is not antimicrobially active; however, further research is required.

Fast removal of cloxacillin residues after DCT was also found by Pérez et al. [28]. They treated six cows with 500 mg cloxacillin benzathine in the seventh month of pregnancy and analyzed milk samples from the 1st, 3rd, 6th, 9th and 12th milking after calving. Using a detection limit of 10 µg/kg, no cloxacillin residues were found from the first milking after calving. Burmańczuk et al. [27] described a biphasic elimination of cloxacillin from the udder. After intramammary application of 500 mg of cloxacillin benzathine in lactating cows, the compound was eliminated in a fast elimination phase until 60 h after treatment followed by a slow elimination phase until 100% wash out after 6.5 d. In dry cows treated with 600 mg of cloxacillin benzathine, the slow elimination phase ended on d 21 after treatment, followed by fast elimination with total wash out after 35 d. In 7 out of 10 cows, cloxacillin concentrations exceeded 30 µg/kg up to 21 d after treatment. Samples were drawn from 3 d after treatment in the dry period.

As cloxacillin is an anionic molecule with a pKa of 2.7 and pH of around 6.8 in the mammary gland [28], the molecule is expected to be present in the non-dissociated state and thus unable to cross the cellular membrane [40]. A total of 1000 mg of cloxacillin are expected to be absorbed slowly with tissue concentrations remaining low but sufficient to be effective against *Staphylococcus aureus* (MIC_90_ of 0.5 μg/mL; [41]). Hence, the levels of cloxacillin found in the mammary gland during the dry period can be considered therapeutic even though the drug would have practically disappeared in the milk by the time milking is resumed [28]. However, commercial dairy producers sporadically report cloxacillin residues in milk samples taken after antibiotic dry off [42]. A possible explanation is a violation of the withdrawal period due to incorrect application of the product guidelines or accidental use of milk from treated cows [43]. Alternatively, unexpected residues in individual cows caused by slow removal could be responsible.

In agreement with our data, previous studies [27,41] reported a high variation in cloxacillin removal between individual cows (Table 1 and Table 2). In a small number of cows, the excretion of cloxacillin was slower than in others, which might explain rare cases of antibiotic residues in milk beyond 5 DIM after DCT [43]. Moreover, in the present study, individual cows had higher concentrations of cloxacillin and cloxacillin benzathine (1:1) at later milking despite showing initially low residues. However, no concentrations above the MRL were detected after the withdrawal period of 5 d. As in the present study, we observed an association of cloxacillin benzathine (1:1) excretion with milk protein and lactose concentration, changing milk composition might contribute to irregular residue depletion.

The day of DCT relative to calving had a significant effect on cloxacillin and cloxacillin benzathine (1:1) concentrations in milk (Table 1 and Table 2). Shorter treatment to calving intervals was associated with higher concentrations of both compounds, which is in agreement with an earlier report from our group on a cefquinome DCT (150 mg) applied on d 21, 14, and 7 before the estimated calving date [31]. They were able to demonstrate the effect of treatment day until the 37th milking after calving and found cefquinome residues above the MRL until d 21 after calving.

In the present study, the lactose content of milk was associated with cloxacillin benzathine (1:1) excretion. Interestingly, Bachmann et al. [31] found the same effect of lactose on cefquinome in the first milking after calving. Andrew et al. [44] reported an influence of milk composition on the accuracy of antibiotic residue tests and assumed that milk ingredients may interact with antibiotic residues. One might speculate that lactose can bind cloxacillin or cloxacillin benzathine (1:1) and thus increase the concentration detected when lactose excretion is high. Further research is warranted to elucidate this phenomenon.

We also found traces of cloxacillin and cloxacillin benzathine 1:1 in the untreated udder quarters (Table 1 and Table 2). Possible explanations include contamination during sample collection or contamination during the processing of samples before cloxacillin measurement. However, because the cloxacillin and cloxacillin benzathine (1:1) concentrations diminish with each milking, this seems unlikely. These concentrations might also be attributed to carry-over effects as observed by Sanford et al. [45]. These authors found cloxacillin residues in 25% of untreated udder quarters located ipsilaterally or diagonally to quarters treated with cloxacillin DCT. The mean cloxacillin concentration in untreated quarters was 0.006 μg/mL (median, 0 μg/mL; range, 0 to 0.19 μg/mL). They presumed that transfer was facilitated by systematic circulation. A similar observation has been made for untreated quarters neighboring those treated with cefquinome [31,46].

### Study Limitations

Cows in this study were dried off nine weeks before calving. They received antibiotic DCT 28, 21, and 14 d before the expected calving date. By the time of DCT udder involution was completed. Drug removal may be different when DCT and dry off are conducted at the same time.

Cows included in this study were housed at the Clinic for Animal Reproduction. As milking was conducted twice daily, the results cannot necessarily be transferred to herds that are milked one to three times daily. All cows included in this study were healthy. It is possible that cloxacillin excretion differs in cows with diseases. Moreover, all cows enrolled were at the end of their first lactation. Whether the results obtained from this study can be applied to cows in ≥2nd lactation remains speculative.

## 5. Conclusions

In the 15 cows enrolled in this study, three udder quarters were treated with 1000 mg cloxacillin benzathine (2:1) on d 252, d 259, and d 266 of gestation. Although the interval between treatment and calving was considerably short, all cows showed a cloxacillin concentration below the MRL of 30 µg/kg after 5 d. According to the HPLC-MS/MS chromatograms two different cloxacillin metabolites were detected, i.e., cloxacillin and cloxacillin benzathine (1:1). While in udder quarters of G14d, the cloxacillin concentration was below the MRL on the 5th DIM, the cloxacillin concentrations of G21d and G28d were already below the MRL at 1st DIM. The cloxacillin benzathine (1:1) concentration in the milk of G28d, G21d, and G14d fell below 30 µg/kg on the fifth, third, and fifth DIM, respectively.

The assigned withdrawal period for the cloxacillin DCT used in this study was 40 d for cows treated less than 35 d before calving. Our results provide evidence that the length of the dry period has an influence on cloxacillin concentration in milk after calving. However, the elimination of cloxacillin after calving was fast. The concentration of cloxacillin metabolites in milk was below the MRL of 30 µg/kg at 5 DIM, even if the interval from treatment to calving was shorter than 14 d.

## Figures and Tables

**Figure 1 animals-13-02558-f001:**
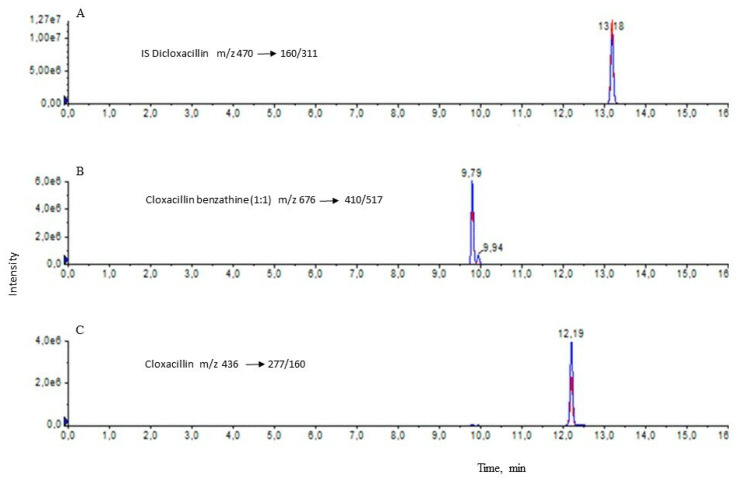
MS/MS chromatogram of spiked sample foremilk with the internal standard 500 µg/kg dicloxacillin (panel (**A**); 13.18 min), 100 µg/kg cloxacillin benzathine 1:1 (panel (**B**); 9.79 min), and 100 µg/kg cloxacillin (panel (**C**); 12.19 min).

**Figure 2 animals-13-02558-f002:**
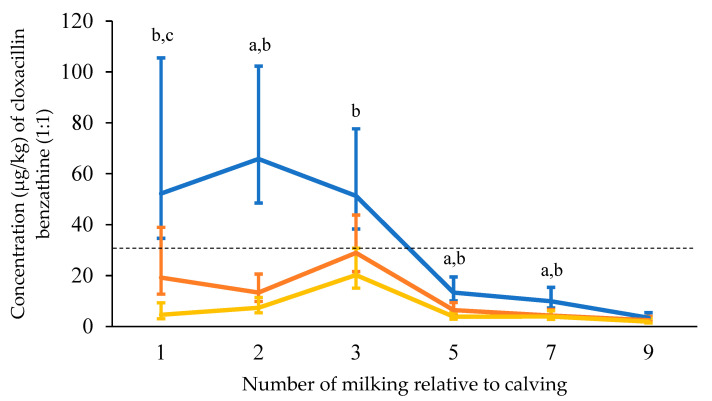
Mean cloxacillin benzathine (1:1) residue concentrations in foremilk over the first 9 milkings after calving. Milk of udder quarters was grouped based on the interval between treatment and calving into 6 to 17 d (G14d; *n* = 17; blue), 18 to 24 d (G21d; *n* = 15; orange), and 25 to 32 d (G28d; *n* = 13; yellow). Least square estimates (mean ± 95% CI) were used from the repeated measures ANOVA. Cloxacillin benzathine (1:1) residue concentration was affected by the number of milking relative to calving (*p* < 0.001), interval between treatment and calving (*p* < 0.001), protein content of the milk (*p* = 0.015), lactose content of the milk (*p* < 0.001), and the number of milking relative to calving by interval from treatment to calving (*p* < 0.066). The dashed line indicates the maximum residue limit. Pairwise comparisons among groups with Bonferroni’s corrected *p* < 0.05 are marked with the following letters: a = G14d vs. G21d, b = G14d vs. G28d; c = G21d vs. G28d.

**Table 1 animals-13-02558-t001:** Mean, median, and range of cloxacillin concentration (µg/kg) in foremilk for the first 9 milkings after calving by treatment group.

	Cloxacillin Concentration in µg/kg											
Number of Milking	G14d ^1^			G21d ^2^			G28d ^3^			Untreated Udder Quarter ^4^		
	Mean	Median	Range	Mean	Median	Range	Mean	Median	Range	Mean	Median	Range
1	93.64	0.87	0.00–1203.12	1.12	0.01	0.00–10.94	1.55	0.01	0.00–5.76	0.01	0.01	0.00–0.06
2	77.77	0.09	0.00–1132.94	1.19	0.02	0.00–12.14	3.55	0.01	0.00–34.70	0.03	0.02	0.00–0.11
3	64.33	0.02	0.00–1026.19	5.39	0.09	0.00–58.15	0.63	0.04	0.01–5.83	0.05	0.01	0.00–0.32
5	14.94	0.03	0.00–238.46	0.37	0.06	0.00–4.64	0.41	0.03	0.00–2.61	0.03	0.02	0.00–0.07
7	6.31	0.00	0.00–83.22	0.69	0.00	0.00–6.65	2.14	0.00	0.00–19.29	0.04	0.00	0.00–0.54
9	2.87	0.00	0.00–41.25	2.18	0.00	0.00–28.16	0.55	0.00	0.00–3.12	0.00	0.00	0.00–0.01

^1^ Cloxacillin concentration in milk of udder quarters (*n* = 17) with an interval of 6 to 17 d between treatment and calving. ^2^ Cloxacillin concentration in milk of udder quarters (*n* = 15) with an interval of 18 to 24 d between treatment and calving. ^3^ Cloxacillin concentration in milk of udder quarters (*n* = 13) with an interval of 25 to 32 d between treatment and calving. ^4^ Cloxacillin concentration in milk of udder quarters (*n* = 15) that did not receive any dry cow treatment before calving.

**Table 2 animals-13-02558-t002:** Mean, median, and range of cloxacillin benzathine (1:1) concentration (µg/kg) in foremilk for the first 9 milkings after calving by treatment group.

	Cloxacillin Benzathine (1:1) Concentration in µg/kg											
Number of Milking	G14d ^1^			G21d ^2^			G28d ^3^			Untreated Udder Quarter ^4^		
	Mean	Median	Range	Mean	Median	Range	Mean	Median	Range	Mean	Median	Range
1	731.27	216.20	39.08–3599.80	220.22	50.83	5.05–1142.09	407.49	12.52	0.12–4751.66	0.22	0.03	0.00–1.05
2	347.55	101.13	5.27–1869.67	106.09	28.95	2.63–572.06	239.59	13.13	0.78–2714.07	0.07	0.00	0.00–0.35
3	154.83	26.05	2.00–1405.05	37.81	20.43	0.00–151.57	50.15	18.95	2.79–343.80	2.78	1.90	0.00–19.21
5	34.67	6.41	0.31–436.96	4.95	2.87	0.00–18.04	8.55	1.20	0.30–79.97	0.12	0.01	0.00–1.72
7	10.49	2.51	0.00–90.50	3.05	2.33	0.00–8.28	5.74	1.01	0.00–50.34	0.24	0.00	0.00–2.30
9	2.21	0.74	0.00–12.23	3.64	0.90	0.00–36.81	1.48	0.79	0.00–7.15	0.00	0.00	0.00–0.05

^1^ Cloxacillin benzathine (1:1) concentration in milk of udder quarters (*n* = 17) with an interval of 6 to 17 d between treatment and calving. ^2^ Cloxacillin benzathine (1:1) concentration in milk of udder quarters (*n* = 15) with an interval of 18 to 24 d between treatment and calving. ^3^ Cloxacillin benzathine (1:1) concentration in milk of udder quarters (*n* = 13) with an interval of 25 to 32 d between treatment and calving. ^4^ Cloxacillin benzathine (1:1) concentration in milk of udder quarters (*n* = 15) that did not receive any dry cow treatment before calving.

## Data Availability

Due to privacy reasons, data are cannot made public, but can made available after special permission.

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
