# Peer review of "Residue Concentrations of Cloxacillin in Milk after Intramammary Dry Cow Treatment Considering Dry Period Length"

_animals, 2023, doi:10.3390/ani13162558_

Round 1
Reviewer 1 Report
The authors' studies are aimed at studying the residue concentrations of cloxacillin in milk after intramammary dry cow treatment considering dry period length.
However, there are some comments (the file with the comments is attached to the form).
1. Line 24-42 (abstract). Understandable writing of the summary of the conducted research is required! At the end of the abstract, it is necessary to give a generalizing conclusion or assessment of the results obtained.
2. Line 95-97 et seq. It should be clarified that residual amounts of specific compounds of the drug are being exhausted!
3. Line 103-105 et seq. The studies were carried out with violations. The control group was not chosen correctly. In this case, the fourth quarter of the udder cannot happen as a control. At the same time, the remaining quarters of the udder are treated with the drug! The animal has a single blood flow system and therefore the drug will also enter the fourth quarter of the udder. And in this case, the results described as controls have no scientific significance!
4. No description of study drug. This is a drug that is undergoing clinical study. Or preparation, which is in constant practice of application. In the latter case, he has already passed all the studies. How correct is it to conduct repeated studies of the drug, the results of which are already known?
5. Line 203-204. On what basis was the study of residual concentrations of specific compounds chosen?
6. Table 3. No unit of measurement results!
7. Line 393-397. My assumptions about control are confirmed in this text. Therefore, changes should be made to the text of the manuscript. And write that the study was conducted without control. But with the study of the distribution of the drug in the body. At the same time, the concentration of residual substances in the untreated quarter of the udder was evaluated!
8. Conclusion. There is no final assessment or conclusions based on the results!
The manuscript requires editing taking into account the comments.

The text of the manuscript is written in English. The text contains minor errors and technical typos. This needs some editing.
Author Response
Reviewer 1
The authors' studies are aimed at studying the residue concentrations of cloxacillin in milk after intramammary dry cow treatment considering dry period length.
However, there are some comments (the file with the comments is attached to the form).
- Line 24-42 (abstract). Understandable writing of the summary of the conducted research is required! At the end of the abstract, it is necessary to give a generalizing conclusion or assessment of the results obtained.
- Thank you for your comment. The Introduction and the Conclusion of the Abstract have been rephrased as follows:
L 24-27. “Dry cow treatment with an intramammary antibiotic is recommended to re-duce the risk of mastitis at the beginning of next lactation. The dry period may be shortened unintentionally, affecting residue depletion and the time when residues reach concentrations below the maximum residue limit (MRL).”
L 40-42. “Shortening the dry period affects the residue depletion after dry cow treatment with cloxacillin. The risk of exceeding the MRL, however, seems low, even with dry periods shorter than 14 d.”
- Line 95-97 et seq. It should be clarified that residual amounts of specific compounds of the drug are being exhausted!
- This comment seems to refer to our hypothesis. We are uncertain what is meant by the reviewer. Please clarify.
- Line 103-105 et seq. The studies were carried out with violations. The control group was not chosen correctly. In this case, the fourth quarter of the udder cannot happen as a control. At the same time, the remaining quarters of the udder are treated with the drug! The animal has a single blood flow system and therefore the drug will also enter the fourth quarter of the udder. And in this case, the results described as controls have no scientific significance!
- As described in the manuscript, it seems to be common practice to consider individual udder quarters as independent study units (See Schukken et al. (1993), doi:10.3168/jds.S0022-0302(93)77632-8; Lindmark-Månsson et al. (2006), doi:10.1016/j.idairyj.2005.07.003; Bachmann et al. (2018), doi:10.3168/jds.2017-13826; Sanford et al. (2006), doi:10.2460/ajvr.67.7.1140). You are, however, right. We included an untreated udder quarter to evaluate whether the dry cow antibiotic can be transferred into another udder quarter. Therefore the term “negative control” is misleading and was replaced by “untreated udder quarter” throughout the manuscript.
- No description of study drug. This is a drug that is undergoing clinical study. Or preparation, which is in constant practice of application. In the latter case, he has already passed all the studies. How correct is it to conduct repeated studies of the drug, the results of which are already known?
- The veterinary medicinal product used in this study is an authorized drug. The authorisation number has been added to the text to make this clear (see below). However, the present study is in not a repetition of residue depletion studies conducted for the authorisation procedure. Those residue depletion studies need to follow the rules described in the applicable EMA guideline (“Determination of withdrawal periods for milk”, EMA/CVMP/SWP/735418 Rev1*) which is a complety different study design using animals with dry periods as similar as possible. The study described in this paper was conducted with the intention to determine the effect of different dry cow periods after DCT on cloxacillin residue concentrations in milk after calving.
L147-151. “The treatment protocol was initiated using an antibiotic DCT consisting of 1,000 mg cloxacillin benzathine 2:1 (Cloxacillin-Benzathin 1,000 mg T.S., CP-Pharma, Burgdorf, Germany; compound consisting of two molecules cloxacillin and one molecule of benzathine; authorization number 6232.00.00).”
- Line 203-204. On what basis was the study of residual concentrations of specific compounds chosen?
- Initially, we only evaluated the concentration of cloxacillin. As we detected a considerable signal at 9.79 min in the chromatogramms that was identified as cloxacillin benzathine 1:1, we also quantified the concentration of cloxacillin benzathine 1:1. This is described in the following two paragraphs:
Material and Methods:L 205-212. “During preliminary investigations of study samples, a considerable signal in the chromatograms was observed. The signal was identified as cloxacillin benzathine (1:1) by comparison with a reference standard and shows another retention time than cloxacillin in the chromatogram. Cloxacillin benzathine (1:1) is a compound in which one molecule of cloxacillin is linked to one molecule of benzathine. Thus, we decided to analyze cloxacillin and cloxacillin benzathine (1:1) separately.”
Results: L 321-328. “The HPLC-MS/MS revealed a peak at 9.79 min and a second peak at 12.19 min. Using cloxacillin and cloxacillin benzathine (1:1) spiked calibration samples, it was confirmed that the peak detected at 9.79 min referred to cloxacillin benzathine (1:1) and the peak detected at 12.19 min referred to cloxacillin. A limit of quantification (LOQ) of 0.1 µg/kg was determined for cloxacillin and cloxacillin benzathine (1:1) each. As we observed different excretion kinetics for cloxacillin and cloxacillin benzathine (1:1), results are reported separately.”
- Table 3. No unit of measurement results!
- As the major findings are described within the text and the estimates are difficult to interpret (logarithmic scale, interaction), this table added little value to the reader. Table 3. has therefore been removed.
- Line 393-397. My assumptions about control are confirmed in this text. Therefore, changes should be made to the text of the manuscript. And write that the study was conducted without control. But with the study of the distribution of the drug in the body. At the same time, the concentration of residual substances in the untreated quarter of the udder was evaluated!
- See comment above.
- Conclusion. There is no final assessment or conclusions based on the results!
- The Conclusion has been rephrased as follows.
L 498-512: “In the 15 cows enrolled in this study, three udder quarters were treated with 1,000 mg cloxacillin benzathine (2:1) on d 252, d 259, and d 266 of gestation. Although the interval between treatment and calving were considerable short, all cows showed a cloxacillin concentration below the MRL of 30 µg/kg after 5 d. According to the HPLC-MS/MS chro-matograms 2 different cloxacillin metabolites were detected i.e. cloxacillin and cloxacillin benzathine (1:1). While in udder quarters of G14d the cloxacillin concentration was below the MRL on the 5th DIM, the cloxacillin concentrations of G21d and G28d were al-ready below the MRL at 1st DIM. The cloxacillin benzathine (1:1) concentration in the milk of G28d, G21d, and G14d fell below 30 µg/kg on the 5th, 3rd and 5th DIM, respectively. The assigned withdrawal period for the cloxacillin DCT used in this study was 40 d for cows treated less than 35 d before calving. Our results provide evidence that the length of dry period has an influence on cloxacillin concentration in milk after calving. However, elimination of cloxacillin after calving was fast. The concentration of cloxacillin metabo-lites in milk was below the MRL of 30 µg/kg at 5 DIM, even if the interval from treatment to calving was shorter than 14 d.
The manuscript requires editing taking into account the comments.
- Thank you for your comments. We hope that all suggestions made by the reviewer have been satisfactorily answered.
- Chapter 3.2 Residue Depletion – Cloxacillin: Please mention „Table 1“ as a description, where the mentioned values are displayed.
- A reference to table 1 has been included.
L 332-334. “In G21d and G28d, however, a concentration above the MRL was detected on d 3 (58.15 µg/kg), and d 2 (34.70 µg/kg), respectively (Table 1).”
- Line 344: Please delete „in“.
- Thank you. Deleted.
- Please try to explain, why individual cows had higher concentrations of cloxacillin and cloxacillin benzathine (1:1) at a later milking despite showing initially low residues (lines 457-459), what results in increased means on milking number 3 for G21d and on milking numbers 2 and 7 for G28d (Table 1).
- The following lines were added to this abstract.
L 460-463. “As in the present study we observed an association of cloxacillin benzathine (1:1) excretion with milk protein and lactose concentration, changing milk composition might con-tribute to irregular residue depletion.”
- I believe there is a reference missing at the end of the first sentence of the introduction: „… and to improve cure rate in cows with subclinical mastitis (?)“.
- Rephrased.
- Abstract, lines 36-37: „All cows showed a cloxacillin concentration below the MRL of 30 µg/kg after 5 d.“ Here you speak about the mean values. In lines 38-40: „The cloxacillin benzathine (1:1) concentration in the milk of G28d, G21d, and G14d fell below 30 µg/kg on the ninth, fifth and ninth milking, respectively.“ Here you speak about the maximum values of the range. This is confusing. You maybe consider a reformulation of the first sentence, e.g.: „Across all milkings a mean cloxacillin concentration below 30 µg/kg was found from day 5.“.
- All data refer to the maximum concentrations. In the first sentence we wanted to point out, that the concentrations of both cloxacillin metabolites (i.e. cloxacillin and cloxacillin benzathine [1:1]) were below the MRL after the 5th DIM in all cows. Then we spezified the individual DIM for the three different groups when maximum cloxacillin and cloxacillin benzathine (1:1) concentrations fell below the MRL. To make this clearer, the paragraph has been rephrased as follows.
L 34-39. “All cows showed a cloxacillin and cloxacillin benzathine (1:1) concentration be-low the MRL of 30 µg/kg after 5 d. In udder quarters of G21d and G28d, the cloxacillin concentration was already below the MRL at 1st milking after calving. The cloxacillin benzathine (1:1) concentration in the milk of G28d, G21d, and G14d fell below 30 µg/kg on the 5th, 3rd and 5th DIM, respectively.”

Reviewer 2 Report
The submitted study tries to uncover the residue depletion in milk of cows treated with oxacillin, a commonly used antibiotic for dry cow therapy, in relation to the time of administration pre-calving.
The subject of this study is of prime importance in the scientific field and in the dairy industry, especially considering the great amount of efforts made by authorities to reduce antibiotics usage and improve food quality and safety.
Overall, the work can be considered well made and presented, but nontheless there are some minor corrections to be made.
Line 113-116: this part is not very clear, I suggest to rephrase it.
Line 344: remove "in"
Table 3: remove the superscript "1" from "Variable"
Author Response
The submitted study tries to uncover the residue depletion in milk of cows treated with oxacillin, a commonly used antibiotic for dry cow therapy, in relation to the time of administration pre-calving.
The subject of this study is of prime importance in the scientific field and in the dairy industry, especially considering the great amount of efforts made by authorities to reduce antibiotics usage and improve food quality and safety.
Overall, the work can be considered well made and presented, but nontheless there are some minor corrections to be made.
Line 113-116: this part is not very clear, I suggest to rephrase it.
Thank you for your comment. The sentence has been rephrased as follows.
“Due to the comprehensive sampling scheme, the study was conducted in 3 consecutive replicates. While 5 cows were enrolled in the 1st replicate, 6 cows were enrolled in each of the remaining 2 replicates.”
Line 344: remove "in"
- Removed.
Table 3: remove the superscript "1" from "Variable"
- Thank you for reading carefully. The whole table has been removed as it added little value to the paper.

Reviewer 3 Report
Dear authors,
I found your work to be relevant for research and well structured. However, I have some minor comments to make.
Minor comments:
1. Please check the document for consistency, e.g. when numbers are given (5 or five, e.g. line 22)). Also there is some inconsistency in use of spaces or no spaces between values and units (30% or 30 %).
2. In line 119 the abbreviation „DHIA“ is used. Please provide the definition of this abbreviation.
3. Line 278: Please delete the „,“ after Dohoo et al.
4. Chapter 3.2 Residue Depletion – Cloxacillin: Please mention „Table 1“ as a description, where the mentioned values are displayed.
5. Line 344: Please delete „in“.
6. Please try to explain, why individual cows had higher concentrations of cloxacillin and cloxacillin benzathine (1:1) at a later milking despite showing initially low residues (lines 457-459), what results in increased means on milking number 3 for G21d and on milking numbers 2 and 7 for G28d (Table 1).
7. I believe there is a reference missing at the end of the first sentence of the introduction: „… and to improve cure rate in cows with subclinical mastitis (?)“.
8. Abstract, lines 36-37: „All cows showed a cloxacillin concentration below the MRL of 30 µg/kg after 5 d.“ Here you speak about the mean values. In lines 38-40: „The cloxacillin benzathine (1:1) concentration in the milk of G28d, G21d, and G14d fell below 30 µg/kg on the ninth, fifth and ninth milking, respectively.“ Here you speak about the maximum values of the range. This is confusing. You maybe consider a reformulation of the first sentence, e.g.: „Across all milkings a mean cloxacillin concentration below 30 µg/kg was found from day 5.“.
Author Response
Thank you for carefully reviewing our manuscript. We highly appreciate your effort and time to contribute in this reviewing process.
Please check the document for consistency, e.g. when numbers are given (5 or five, e.g. line 22)). Also there is some inconsistency in use of spaces or no spaces between values and units (30% or 30 %).
- Thank you for reading carefully. We switched to Arabic numbers throughout the manuscript. Spaces between numbers an “%” have been removed, whereas a space was kept in front of all other units.
- In line 119 the abbreviation „DHIA“ is used. Please provide the definition of this abbreviation.
- The sentence has been rephrased as follows:
L 114-120. “Cows were eligible for inclusion if they were (1) Holstein Friesian cows, (2) at the end of their first lactation, around 1 wk before dry off, (3) had no case of clinical mastitis during their first lactation, (4) had a somatic cell count below 100,000 cells/ml in their last 3 dairy herd improvement asso-ciation (DHIA) equivalent tests (monthly test for milk yield, milk compo-nents and somatic cell count), and (5) did not receive antibiotic DCT before the start of the study.”
- Line 278: Please delete the „,“ after Dohoo et al.
- Thank you. Removed.

Round 2
Reviewer 1 Report
The authors have made corrections to the manuscript.
A significant part of the comments has been corrected.
However, the following may be noted.
1. Line 24-26. It is not clear what kind of depletion of the remains in question. It is necessary to write clearly because this is an abstract and it must be clear!
2. Based on the study, the meaning of the study of the untreated quarter of the udder is not clear! What was the purpose of this study? The text of the manuscript does not make this clear. The authors studied the level of residues of the metabolites of the drug when administered in 1-3 quarters of the udder. It is clear! And why do we study the levels of metabolites in the 4th quarter? The purpose of this is not clear!
3. The purpose of the study is not written in the introduction or in the materials and methods. This makes it difficult to understand the study!
4. Line 102-105. What is the purpose of this mention? If we do not have control and an independent group of cows, then there is no point in this mention! In the authors, all groups of animals were not separate and isolated in the treatment with the drug! The animal's circulatory system does not allow for the study of different concentrations of the drug in different quarters of the same udder!
5. Clause 3.4. What is the purpose and purpose of the data presented here (at this point)?
Corrections by the authors are required to make a final decision on the manuscript. Or do you need a detailed answer from the authors!
English requires minor editing!
Author Response
Dear reviewer,
thank you again for being so quick and taking your time for reviewing our manuscript. We deeply understand your concerns with the untreated udder quarter, but hope that we could explain, why we planned and did it like this. We are sorry with our lack of understanding of the third comment, but maybe you could clarify this to us.
Thank you again.
- Line 24-26. It is not clear what kind of depletion of the remains in question. It is necessary to write clearly because this is an abstract and it must be clear!
AU: We understand your concern. We added “antibiotic” residue to clarify this issue.
, affecting antibiotic residue depletion
- Based on the study, the meaning of the study of the untreated quarter of the udder is not clear!What was the purpose of this study?The text of the manuscript does not make this clear.The authors studied the level of residues of the metabolites of the drug when administered in 1-3 quarters of the udder.It is clear!And why do we study the levels of metabolites in the 4th quarter?The purpose of this is not clear!
AU: As you pointed out in your first review, the untreated quarter cannot serve as a control, which we fully agreed with. As this study is not the first of this kind, we followed the study design of the preceding studies for comparability. As in other studies, we found residues of antibiotic in the milk of untreated quarters, which is an important information. In udder health management one quarter is normally assumed as a single unit. The knowledge of antibiotic transfer in untreated quarters is important for preventing antibiotic resistance. This is especially true, as for reduction of use of antibiotics there is discussion of treating only single quarters for DCT units (Robert, A., Bareille, N., Roussel, P., Poutrel, B., Heuchel, V., & Seegers, H. (2006). Interdependence of udder quarters for new intramammary infection during the dry period in cows submitted to selective antibiotic therapy. Journal of dairy research, 73(3), 345-352).
- The purpose of the study is not written in the introduction or in the materials and methods.This makes it difficult to understand the study!
AU: We struggle a bit with this comment. We meant to describe the objective of the study in L92-96 in the introduction;
The objective of the present study was to determine the effect of differ-ent dry cow periods after DCT on cloxacillin residue concentrations in milk after calving. Our hypothesis was that cloxacillin concentrations in milk would exceed the MRL for a prolonged period of time if a short dry period (i.e., < 21 d) is used.
We are sorry for our lack of understanding, but could you please clarify, what kind of information is missing?
- Line 102-105.What is the purpose of this mention?If we do not have control and an independent group of cows, then there is no point in this mention!In the authors, all groups of animals were not separate and isolated in the treatment with the drug!The animal's circulatory system does not allow for the study of different concentrations of the drug in different quarters of the same udder!
AU: We understand your concern. We have mentioned this for two reasons. First, in our scientific community we have very strict regulations in terms of animal experiments. We have to encounter every possible method to reduce the number of animals needed for the experiment. Due to the anatomy of the cow udder, it is generally assumed that the udder quarters are independent, so for us it was feasible to use one quarter as the experimental unit. But as we agree with your argument, that some of the antibiotic will be resorbed in the blood flow (although the formulation of udder treatment is suppose to limit this to a minimum), we left one udder quarter untreated to exactly have a measurement of the amount of transfer to untreated quarters. Our results show, that there is a transfer, but very minimal.
We hope that we now could explain our study design to you and clarify your concerns.
- Clause 3.4.What is the purpose and purpose of the data presented here (at this point)?
AU: We hope we could explain this with answer to remark 4. We found it an important and valuable information for above named reasons. We hope that we could reply to your concerns satisfactory. Thank you again for reviewing this article.
